# Impact of Obesity on Clinical Outcomes of Patients with Intra-Abdominal Hypertension and Abdominal Compartment Syndrome

**DOI:** 10.3390/life13020330

**Published:** 2023-01-24

**Authors:** Swetha Mohan, Zavier Yongxuan Lim, Kai Siang Chan, Vishal G. Shelat

**Affiliations:** 1Lee Kong Chian School of Medicine, Nanyang Technological University, Singapore 308232, Singapore; 2Tan Tock Seng Hospital, Singapore 308433, Singapore

**Keywords:** abdominal compartment syndrome, intra-abdominal hypertension, intra-abdominal pressure, obesity

## Abstract

Intra-abdominal hypertension (IAH) and abdominal compartment syndrome (ACS) are associated with high morbidity and mortality. Obesity may result in increased intra-abdominal pressure (IAP) and affect clinical outcomes of patients with IAH and/or ACS. This study aims to establish the impact of obesity on the clinical outcomes of IAH and ACS patients. A systematic search of Medline, Embase, and Scopus was performed in August 2022. Nine studies comprising 9938 patients were included. There were 65.1% males (n = 6250/9596). Patient demographics, comorbidities, and morbidities were analyzed in correlation with obesity and IAP. Obese patients had a higher risk of IAH (OR 8.5, *p* < 0.001). Obesity was associated with the need for renal replacement therapy, intensive care unit-acquired infections, systemic inflammatory response syndrome, acute respiratory distress syndrome, length of hospital stay, and mortality. This review highlights the lacunae in the existing literature to underpin the direct impact of obesity, independent of obesity-associated comorbidities, on the clinical outcomes of IAH and ACS.

## 1. Introduction

Intra-abdominal hypertension (IAH) and abdominal compartment syndrome (ACS) are clinical manifestations attributed to raised intra-abdominal pressure (IAP) [1]. Though Kron et al. first described characteristics of raised abdominal pressure, it was Fietsam et al. who first coined the term ACS in the context of a ruptured abdominal aortic aneurysm [1,2]. Advances in clinical management have led to increased recognition of these disease spectrums in recent years [3]. The World Society for Abdominal Compartment Syndrome (WSACS) was founded in 2004 to promote research, foster education, and improve survival of patients with IAH and ACS [4]. The WSACS 2013 guidelines defined IAH as a sustained or repeated pathological elevation in IAP ≥12 mmHg with varying severity grades [4]. ACS is defined as sustained IAP >20 mmHg associated with new organ dysfunction or failure. IAP can be measured using various techniques including the placement of catheters and pressure transducers in the stomach, rectum, urinary bladder, inferior vena cava, and peritoneal cavity. The most common technique involves the placement of a catheter into the partially filled urinary bladder, and in patients without previous urinary bladder surgery or pelvic adhesions, this is generally accurate and reliable. WSACS recommends the method described by Kron et al. to assess IAP: this involves instillation of up to 25 mL of saline into the bladder with IAP measured at end-expiration in a supine position with the transducer zeroed and positioned in line with the iliac crest and mid-axilla after at least 30–60 s of saline instillation [4,5].

IAH and ACS are increasingly recognized as complications in intensive care unit (ICU) patients and are feared due to their associated high morbidity and mortality [5]. Their complications and organ dysfunctions are largely due to either decreased abdominal perfusion pressure or the intrinsic or extrinsic compression of organs. Primary causes of IAH include acute pancreatitis, abdominal trauma, abdominal aortic aneurysm rupture, and retroperitoneal hematoma [6]. Secondary causes of IAH include ascites, ileus, intra-abdominal sepsis, and large-volume fluid replacement. To reduce the morbidity and mortality burden of ACS, monitoring of IAP is essential, as clinical care provision could be tweaked to prevent progression of IAH to ACS. In patients with established ACS, non-surgical approaches like nasogastric tube insertion, flatus tube insertion, sedatives, fluid restrictive regimens, diuresis, etc. are useful. In general, surgical management of ACS is reserved for ACS with persistent organ dysfunction refractory to non-operative treatment. 

Pre-existing comorbidities such as obesity exacerbate the effects of the raised IAP, reducing the threshold required for the development of ACS [5]. Obesity is increasing in prevalence, with estimates predicting 38% of the world’s adult population and another 20% will be obese by 2030 [7]. Additionally, studies suggest that IAP correlates with obesity and obesity-related comorbidities such as systemic hypertension, gastroesophageal reflux disease, and obstructive sleep apnea [8]. The potential impact of obesity on IAH and ACS is both direct and indirect from associated obesity-related comorbidities. Hence, this study aims to systematically review literature to determine the impact of obesity on (a) IAP values and (b) clinical outcomes of patients with IAH or ACS.

## 2. Materials and Methods

### 2.1. Study Selection and Search Strategy

A systematic search on Medline, Embase, and Scopus was performed from inception to 18 August 2022 on the impact of obesity on IAP and the clinical outcomes of patients with IAH or ACS. This systematic review was performed in accordance with the PRISMA (Preferred Reporting Items for Systematic Reviews and Meta-Analyses) guidelines [9]. This study was registered at PROSPERO before conduct of the study (CRD42022357580). No ethical approval was required for this study, as it was a review study on published literature. A combination of the following search terms was used: “Intra-abdominal hypertension”, “intra-abdominal pressure”, “abdominal compartment syndrome”, and “obesity” or “body mass index”. The detailed search strategy is appended in Appendix A.

The inclusion criteria were randomized controlled trials (RCTs) or non-RCTs assessing the impact of obesity on IAP and the clinical outcomes of patients with IAH or ACS. Exclusion criteria were (a) patients ≤18 years old, (b) animal studies, (c) irrelevance to obesity, IAP, IAH, or ACS, (d) lack of reporting of outcomes, (e) article type (e.g., non-English articles, reviews, editorials, letters to editors), or (f) lack of availability of full-texts.

After the removal of duplicates, the articles were independently reviewed by two authors (S.M. and Z.Y.L.). Subsequently, the articles included in the screening were assessed in full text to determine if they met the inclusion criteria. Any disagreements were resolved in consultation with the senior author (V.G.S.). The study selection process is shown in the PRISMA flow diagram (Figure 1). Forty-three reports were excluded due to article type (case reports, case-series, reviews, editorials, or letters to editors) or were animal studies.

### 2.2. Data Extraction and Synthesis

Data extraction was performed by two independent authors (S.M. and Z.Y.L.). The following variables were extracted: publication details (name of first author, year of study, study period, country of study, study design), study characteristics (sample size, patient demographics including comorbidities and definition of obesity), and study outcomes. Obesity was defined as per the definitions used in the original studies and was included in Table 1. The primary outcomes were mortality and the length of ICU stay. The secondary outcomes were the need for transfusion (packed cells, fresh frozen plasma (FFP), or platelets), hospital length of stay (LOS), the need for vasopressor, and specific morbidity, e.g., sepsis or systemic inflammatory response syndrome (SIRS). A quantitative synthesis of extracted data using a meta-analysis was not performed because of the small number of studies reporting heterogeneous inclusion criteria and outcome assessments.

### 2.3. Risk of Bias Assessment

Quality assessment was performed by two independent authors (S.M. and Z.Y.L.). The risk of bias was assessed using the modified Newcastle-Ottawa scale for observational studies (Table 2).

## 3. Results

### 3.1. Study Characteristics

The initial search yielded a total of 2167 articles, of which 389 duplicates were removed and the remaining 1778 were screened based on abstracts and titles. Subsequently, 126 articles were assessed for eligibility via full-text screening, of which 117 articles were further excluded (Figure 1). The final systematic review included nine articles [8,10,11,12,13,14,15,16,17].

Our study included 9938 patients with 65.1% males (*n* = 6250/9596). There were three studies on ICU patients [10,12,17], three studies on bariatric surgery patients [8,11,16], two studies on cardiothoracic surgery patients [14,15], and one study on patients with acute pancreatitis (AP) [13]. Table 3 summarizes the study characteristics, definitions of obesity, and patient demographics of included studies. The most commonly used definition of obesity was body mass index (BMI) ≥ 30 kg/m^2^ (*n* = 4 studies). One author included only morbidly obese patients (BMI ≥ 35 kg/m^2^) [11], and three authors did not report the BMI cut-off used to define obesity [8,13,14]. In addition, one author included an upper quartile sagittal abdominal diameter (SAD) > 26 cm as a definition of “abdominal obesity” [12]. The majority (*n* = 7/9) of the studies were prospective cohort studies. IAP was measured intravesically for all included studies. Table 4 summarizes the overall comorbidities reported in individual studies in obese and non-obese patients. 

### 3.2. Obesity and Impact on IAP

Studies classified patients’ weight as obese, overweight, normal, or underweight. For the purpose of this study, we further stratified them into two categories: obese and non-obese (defined as underweight, normal and/or overweight). The differences between definitions of BMI were disregarded, and patients were classified as reported by each author. There were five studies which reported the correlation between obesity and IAP [10,11,12,15,16,17]. All of the studies which reported this showed that obesity was associated with raised IAP, IAH, or both. For instance, Kim et al. reported that patients who were obese (BMI ≥ 30) were more likely to have IAH (odds ratio (OR) 8.5 (95% confidence interval (CI) 2.7–31.9, *p* < 0.001) compared to patients who were not obese [10].

### 3.3. Clinical Outcomes of Patients with IAH Or ACS

None of the studies reported the clinical outcomes of obese versus non-obese patients in those who had ACS due to the low incidence of ACS reported. Hence, we were unable to directly analyse the impact of obesity on outcomes in patients with ACS. However, as obesity is demonstrated to be a risk factor for ACS, we compared the incidence of morbidities between obese and non-obese patients. Table 5 reports the incidence of various morbidities in obese and non-obese patients for included studies. There were two studies which reported ICU LOS. Kim et al. reported obesity as a significant predictor of IAH (OR 8.5, 95% CI: 2.7–31.9), but similar ICU LOS in IAH and non-IAH patients [10]. In contrary, while Malbrain et al. reported that obesity had statistically significant correlation with IAP, the association was poor (R^2^ = 0.0413, *p* < 0.001); however, IAH patients had significantly longer ICU LOS compared to non-IAH patients (16.2 ± 15.2 days vs. 4.2 ± 5.9 days, *p* < 0.001) [18]. This suggests that although increased BMI is indeed associated with increased IAP, the variance of the data cannot be solely attributed to obesity. Hence, other factors affecting IAH must also be considered. IAH was also significantly associated with a higher incidence of sepsis, but LOS and mortality were comparable between IAH and non-IAH patients [10]. Kim et al. also noted a significantly higher incidence of sepsis in IAH patients (*p* = 0.013) [10]. At the same time, Ramser et al. reported higher units of blood, platelet, and fresh frozen plaza transfusions, although no *p*-value could be calculated due to the small sample size [14].

## 4. Discussion

IAH and ACS result in significant morbidity and mortality, and a significant proportion of these patients are obese [11,19]. This systematic review analysed existing literature and illustrated that obesity is associated with increased IAP and comorbidities. Obesity is also associated with need for renal replacement therapy (RRT), ICU-acquired infections, SIRS, acute respiratory distress syndrome (ARDS), LOS, and mortality.

Our study showed that obesity is a risk factor for the development of IAH and ACS after ICU admission [10,12,15,17]. Lambert et al. and Sugerman H et al. also reported that the mean IAP in obese patients was significantly higher compared to non-obese patients [11,16]. The literature has demonstrated that obesity is associated with a higher baseline IAP even without any ongoing pathological disease (i.e., in the context of IAH / ACS) [11]. Frezza et al. reported that every 1 kg/mm^2^ increase in BMI is associated with a 0.07 mmHg increase in IAP [20]. It has been postulated that central obesity results in increased visceral fat and a sphere-like baseline shape of the abdominal cavity with poor stretching capacity, thus resulting in increased IAP [15]. The correlation between obesity and increased IAP may be compounded by obesity-related comorbidities such as systemic hypertension, obstructive sleep apnea (OSA), and type 2 diabetes mellitus (DM) [8,20]. This is supported by the study by Lambert et al., which demonstrated that patients with ≥5 comorbidities had significantly higher mean IAP compared to those with <3 comorbidities (mean IAP approximately 14 cmH_2_O (i.e., 10.3 mmHg) vs. 11 cmH_2_O (i.e., 8.1 mmHg), *p* < 0.005) [11]. This raises the question of whether obesity is a cause of IAH. Smit et al. reported that 39.5% (*n* = 15/38) of obese patients had IAH compared to only 23.6% (*n* = 35/148) of the non-obese patients [15]. Although results were not statistically significant (*p* = 0.667), this represents an absolute difference of 15.9%, which is clinically significant. Hence, prudence is required by clinicians to identify patients with obesity and obesity-related complications which predispose them to raised IAP and potentially pathological IAH due to a higher mean baseline IAP. For example, overzealous fluid resuscitation in an obese patient is more likely to elevate IAP beyond the threshold of IAH due to fluid third spacing, bowel edema, and ileus. Thus, prudence is necessary for clinical management of an obese patient and an awareness that resuscitation should be balanced to avoid IAH and prevent ACS in obese patients. The other consideration would be the timing of the measurement of IAP. Smit M et al. cautioned on the wide range of IAP measurements, and centres should be cautious of using a single measure of IAP [15]. Similarly, the WSACS 2013 guidelines recommend using serial or continuous IAP measurements to manage IAH and ACS [4]. Patients at high risk of developing ACS should have periodic serial IAP measurements every 4–6 h [21]. Serial measurements generate more meaningful trends than single point readings and guide clinical decision-making.

Obesity has been shown to be associated with longer mechanical ventilation and ICU LOS duration in critically ill patients [22]. Longer stays increase risks of venous thromboembolisms, pressure sores, lung injury from prolonged mechanical ventilation, and neuromuscular weakness [23,24,25,26]. These complications have a lifelong impact on a patient’s quality of life and physical reserves [27], thus increasing their susceptibility to poorer outcomes. Additionally, in a cohort of 269 patients with AP, Paduraru et al. reported a higher incidence of RRT in obese compared to non-obese patients (37% vs. 11%, *p* < 0.01) [13]. This could be due to reduced end-organ perfusion secondary to obesity-related baseline-raised IAP. ACS falls under the “severe” spectrum of IAH with a high risk for multi-organ failure due to reduced organ perfusion pressures. Further, crystalloid volume replacement to improve renal perfusion can contribute to third spacing, paralytic ileus, and self-perpetuating multi-organ failure. Indications for RRT include acute kidney injury (AKI) resulting in complications such as hyperkalemia, fluid overload, pulmonary oedema, metabolic acidosis, and uremic encephalopathy [28]. These issues are independently associated with poor clinical outcomes secondary to arrhythmias, pulmonary oedema, and respiratory failure [29,30]. The need for RRT implies the severity of AKI and heralds poor prognosis with a mortality risk of up to 40.7% [28]. This is supported by Paduraru et al., who reported that there was higher mortality in obese patients (44% vs. 27%, *p* < 0.01) [13]. Only the study by Paduraru et al. reported on the need for RRT and mortality in obese vs. non-obese patients. However, a definitive conclusion on the impact of obesity on the need for RRT and increased mortality cannot be drawn from their study. Furthermore, their study was in a specific group of patients with ACS secondary to AP, which is an uncommon event in the natural history of pancreatitis (reported to be about 2%) [31]. Mortality of ACS secondary to AP has been reported to be 49% [32].

In our review, the number of studies reporting clinical outcomes in obese vs. non-obese patients and IAH vs. non-IAH patients is small, and the studies are heterogenous. This reiterates that the morbidity and mortality associated with IAH and ACS are understudied [10]. Blaser et al. reported 28-day and 90-day mortality rates of 67.7% and 75.9%, respectively, in ICU patients with concomitant ACS [33]. Their study also demonstrated that a BMI ≥27 kg/m^2^ was independently associated with the risk of IAH (OR 1.94, 95% CI: 1.17–3.21, *p* = 0.01). However, the authors were unable to establish a causal link between BMI and mortality. This could be due to the small sample of obese patients, as the median BMI of the included study population was 26 (range 23–31). Thus, the impact of BMI on mortality remains to be answered. The impact of obesity on morbidity and mortality in critically ill trauma patients is independent of its association with IAH and ACS [34].

DM was reported to be more common in obese patients (16% vs 0%) [11]. DM is associated with impaired immune function, which may increase the risk of infection; additionally, both obesity and DM increase the risk of thromboembolic events due to a prothrombotic state [35]. However, correlation does not imply causation, and further well-designed studies, such as the use of propensity score matching, should be used to determine the true impact of obesity on outcomes. A 1:2 propensity-matched study (*n* = 1935) by Goh et al. on obese patients who had blunt trauma showed that obesity was associated with higher incidence of venous thromboembolism (OR 2.71, 95% CI: 1.27–5.76) [36].

A meta-analysis including 14 studies (*n* = 15,347 obese patients) showed that obesity (BMI ≥30 kg/m^2^) was associated with prolonged ICU LOS (MD 1.08 days, *p* = 0.009) and prolonged duration of mechanical ventilation (MD 1.48 days, *p* = 0.04) in ICU patients; mortality was, however, similar (RR 1.00, *p* = 0.97). In addition, the theory of the “obesity paradox” has been raised in several studies on ICU patients, where obese patients are paradoxically associated with lower mortality [37,38]. This is further reinforced in a recent meta-analysis on obese patients in the ICU (*n* = 23 studies, 199,421 patients); obesity was associated with lower ICU mortality (OR 0.88, *p* < 0.001), hospital mortality (OR 0.83, *p* < 0.001), and long-term mortality (OR 0.69, *p* < 0.001). It has been postulated that adipose tissues secrete adiponectin, which has anti-inflammatory effects [39]. Obese patients may have a lower grade of inflammatory response, thus resulting in protective effects on mortality [40]. In our review, only one study evaluated the impact of obesity on mortality. Paolini et al. showed that obesity (defined as BMI ≥18.5 kg/m^2^ with a sagittal abdominal diameter (SAD) ≥26 cm) was associated with higher mortality (44% vs. 25.3%, *p* < 0.01) [12]. Additionally, in a subgroup of patients with BMI ≤30 kg/m^2^, SAD ≥26 cm was associated with higher incidence of ACS (9.1% vs. 0%, *p* = 0.001) and ICU mortality (52.3% vs. 23.9%, *p* = 0.001). The distinction between obesity as defined by BMI and by waist circumference requires discussion.

Although obesity is defined using height and weight parameters, the term “abdominal obesity” also appears in literature. Abdominal obesity is defined as a ≥88 cm and ≥102 cm waist circumference in women and men, respectively [41]. Waist circumference allows the assessment of abdominal adiposity; high waist circumference has been associated with all-cause and cardiovascular mortality independent of BMI [42,43]. Only Paolini et al. used other parameters apart from BMI to define obesity [12]. The incidence of ACS was reported to be higher in abdominally obese patients (7.3% vs. 0%, *p* < 0.01). However, clinical outcomes in patients who were abdominally obese were not reported.

As described earlier, IAH is a disease spectrum, with ACS falling at the most extreme end accompanied by end-organ failure. Blaser et al. demonstrated that grade 3 IAH (i.e., IAP 21–25 mmHg) was associated with significantly higher 28-day mortality (OR 9.08, *p* < 0.01) compared to grade 2 IAH (IAP 16–20 mmHg) (OR 2.54, *p* < 0.01) [33]. Hence, more severe IAH is likely to worsen outcomes. The duration of IAH should also be a consideration for further studies. In a study of 100 surgical ICU patients, Gupta et al. reported that the duration of IAH is also an independent predictor of 30-day mortality and is, of note, a more significant clinical factor than the development of IAH, with a mean duration of IAH in survivors of 2.85 days as compared to 8.38 days in non-survivors [44]. Similar findings were noted in a study by Kyoung and Hong, where IAH duration was an independent predictor of 60-day mortality with OR 1.196 (*p* = 0.014) [45]. This could be because longer IAH duration predisposes the patient to longer duration of tissue hypoperfusion and cellular hypoxia, and this is associated with a longer ICU LOS and its respective complications. However, retrospective studies cannot establish causation from association, and multicentre prospective studies are warranted to generate high-quality evidence.

There was also a significantly higher Acute Physiologic Assessment and Chronic Health Evaluation (APACHE) II score for patients with a raised IAP at 23.3, as compared to 14.3 for normal-IAP patients (*p* < 0.001). Kim et al. also reported a higher APACHE II and III scores for patients with IAH (*p* = 0.013 and 0.027, respectively) [10]. Components of the APACHE II score include mean arterial pressure, heart rate, and temperature, among others. Based on the above evidence that patients with IAH tend to have higher morbidity, it is a valid argument that the APACHE II parameters mentioned above are more unreliable in response to the physiological stress on the body [46]. Despite this, there was no significant difference in the need for RRT or vasopressors. ICU LOS also had no difference with a mean of 7.5 days for patients with IAH, as compared with 6.3 days for patients without IAH (*p* = 0.33) [10]. This could be attributed to the fact that the APACHE II and III parameters are very nonspecific and hence cannot be validated as an appropriate scoring system in IAH and ACS patients.

This study has its limitations. Firstly, quantitative analysis with meta-analysis could not be performed given the small number of studies with heterogeneous samples (e.g., ICU patients, patients with AP). Although IAH and ACS are not uncommon in critically ill patients, the studies included did not compare outcomes of obese vs. non-obese patients in IAH and ACS. A comparison of outcomes was made between obese vs. non-obese patients, and patients with IAH or ACS vs. no IAH or ACS. Results were extrapolated to identify any direct correlation between obesity and outcomes in IAH and ACS (i.e., where obesity results in raised IAP and IAH, and obese patients had worse outcomes, and IAH patients had worse outcomes). However, this may not imply direct causation. There was also limited reporting of comorbidities in included studies, which limited interpretation of the impact of obesity-related comorbidities on outcomes. Definitions of obesity were also heterogenous; additionally, Asian cut-off values for obesity are different and should be used for studies conducted on Asian populations. Lack of reporting on exact BMI cut-offs by three authors [8,13,14] and the inclusion of a study reporting exclusively on the morbidly obese population [11] also reduces the generalizability of the results of our study.

## 5. Conclusions

Obesity is associated with raised IAP, which may predispose patients to IAH. Obese patients are also predisposed to higher morbidity, such as acute kidney injury, with an increased hospital length of stay. This study highlights the lacunae in evidence on the research of obesity in IAH and ACS and warrants further well-designed prospective large sample studies.

## Figures and Tables

**Figure 1 life-13-00330-f001:**
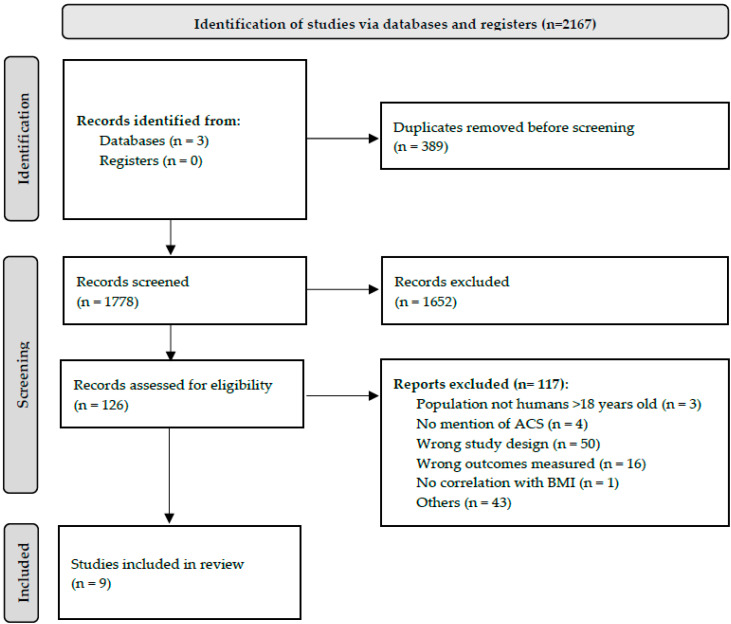
PRISMA Flowchart of Data Screening Process.

**Table 1 life-13-00330-t001:** Summary of study design, inclusion criteria and types of outcomes reported.

No.	First Author	Year	Study Design	Sample Size	Patient Population	Outcomes Reported
1	I. B. Kim [10]	2012	Prospective	100	ICU patients > 18 yo	Survival
2	D. M. Lambert [11]	2005	Prospective	49	Morbidly obese undergoing open Roux-en-Y gastric bypass	Osteoarthropathy, Gallbladder disease, OSA, GERD, HTN, Abdominal hernia, T2DM, HLD, Heart disease
3	J. M. Paolini [12]	2010	Prospective	403	ICU patients	Septic shock, Sepsis, CVS pathology, Neurologic pathology, Haemorrhagic shock, Trauma, ARDS, ICU mortality, RRT, ICU-acquired infection, Acute coronary syndrome, Thrombophlebitis or PE
4	D.N. Pãduraru [13]	2016	Retrospective	269	Acute pancreatitis adult patients	Length of Stay, Mortality
5	M. Ramser [14]	2021	Prospective	4128	Post-cardiac surgery	HTN, Smoking, Dyslipidaemia, Units of blood transfused, Units of FFP transfused, Units of platelets transfused, Mortality
6	M. Smit [15]	2016	Prospective	186	Cardiothoracic surgery patients	COPD, Chronic cardiovascular insufficiency, Immunological insufficiency, Metastasized neoplasm, Respiratory insufficiency, New AKI, Haematological malignancy, New confirmed infection, Vasopressor in first 24 h of ICU, APACHE IV score, LOS in ICU, Reintubation rate
7	H. Sugerman [16]	1997	Prospective	89	84 Bariatric surgery, 5 colectomy w/ ileoanal anastomosis for ulcerative colitis	Hypoventilation, GERD, Venous stasis, Stress incontinence, Incisional hernia, HTN, T2DM
8	J. E. Varela [8]	2009	Prospective	62	Morbidly obese patients who underwent laparoscopic gastric bypass or adjustable gastric banding	Systemic HTN, T2DM, OSA, GERD, Urinary stress incontinence, Lower extremity edema, Abdominal wall hernia
9	P. Wacharasint [17]	2016	Retrospective	4652	SICUs in Thailand	SIRS, New Infection, ARDS, 28-Day Mortality

AKI: Acute kidney injury; APACHE: Acute Physiology and Chronic Health Evaluation; ARDS: Acute respiratory distress syndrome; BMI: Body mass index; COPD: Chronic obstructive pulmonary disease; CVS: Cardiovascular; FFP: Fresh frozen plasma; GERD: Gastro-oesophageal reflux disease; HLD: Hyperlipidemia; HTN: Hypertension; ICU: Intensive care unit; LOS: Length of stay; OSA: Obstructive sleep apnoea; PE: Pulmonary embolism; RRT: Renal replacement therapy; SICU: Surgical intensive care unit; SIRS: Systematic inflammatory response syndrome; T2DM: Type 2 diabetes mellitus.

**Table 2 life-13-00330-t002:** Risk of bias assessment using the Modified Newcastle-Ottawa Scale.

			Selection				Comparability	Outcome			
No.	Year of Study	First Author	Representation of the Exposed Cohort	Selection of Non Exposed Cohort	Ascertainment of Exposure	Demonstration that Outcome of Interest was Not Present at Start of Study	Comparability of Cohorts on the Basis of the Design or Analysis (2 Points)	Assessment of Outcome	Was Follow Up Long Enough for Outcomes to Occur?	Adequacy of Follow Up Cohorts	Total Score^*^
1	2012	I. B. Kim [10]	1	1	1	1	2	1	1	0	8
2	2005	D. M. Lambert [11]	1	1	1	1	2	1	1	0	8
3	2010	J. M. Paolini [12]	1	1	1	1	2	1	1	0	8
4	2016	D.N. Pãduraru [13]	1	1	1	1	2	1	1	0	8
5	2021	M. Ramser [14]	1	1	1	1	2	1	1	0	8
6	2016	M. Smit [15]	1	1	1	1	2	1	1	0	8
7	1997	H. Sugerman [16]	1	1	1	1	2	1	1	0	8
8	2009	J. E. Varela [8]	1	1	1	1	2	1	1	0	8
9	2016	P. Wacharasint [17]	1	1	1	1	2	1	1	1	9

* Quality score of <3: Low quality of evidence, 3–6: Moderate quality of evidence, ≥7: High quality of evidence.

**Table 3 life-13-00330-t003:** Patient demographics of all included studies and respective definitions of obesity.

No.	First Author	Year	Definition of obesity	Sample size	Mean BMI (kg/m^2^)	Incidence of IAH	Mean IAP (cmH_2_O)
1	I. B. Kim [10]	2012	Overweight: BMI ≥ 25 Obese: BMI ≥ 30	Overweight/Obese: 55/100 (55) Non-overweight/Obese: 45/100 (45)	Survivors: BMI 27.7 ± 8.1 Non-survivors: BMI 26.6 ± 7.2 *p* = 0.61	Overweight/Obese: 30/42 (71.4) Non-overweight/Obese: 33/58 (56.9) Obesity on IAH: OR 8.5 (2.7–31.9), *p* < 0.001	Maximum IAP: Survivors: 11.2 ± 5.7 Non-survivors: 11.9 ± 3.7 *p* = 0.60 Mean IAP: Survivors: 10.3 ± 5.5 Non-survivors: 10.6 ± 3.6 *p* = 0.79
2	D. M. Lambert [11]	2005	Morbidly obese: BMI ≥ 35	Morbidly obese: 91.8 (45/49) Control: 8.1 (4/49)	Morbidly obese: 55 ± 2 Control: 26 ± 3 *p* < 0.0001	NR	Morbidly obese: 12 ± 0.7 Control: 0 ± 1.2 ***p* < 0.001**
3	J. M. Paolini [12]	2010	Overweight: BMI ≥ 30 Underweight: BMI < 18.5 Abdominally obese: Upper quartile of SAD (>26 cm)	Underweight: 4.22 (17/403) Control: 68.7 (277/403) Abdominally obese: 27.0 (109/403)	NR	Underweight^*^: 0/17 (0) Control^*^: 1/277 (0.4) Abdominally obese^*^: 8/109 (7.3) ***p* < 0.01** BMI ≤ 30 + SAD ≤ 26 cm^*^: 0/247 (0) BMI ≤ 30 + SAD ≥ 26 cm^*^: 4/44 (9.1)	NR
4	D.N. Pãduraru [13]	2016	NR	NR	NR	102/269 (37.9)	NR
5	M. Ramser [14]	2021	NR	NR	NR	42/4086 (1.0)^*^	
6	M. Smit [15]	2016	Normal weight: BMI < 25 Overweight: BMI 25–29.9 Obese: BMI ≥ 30	Obese: 38/186 (20.4) Non-obese: 148/186 (79.6)	NR	Obese: 15/38 (39.5) Non-obese: 35/148 (23.6)	NR
7	H. Sugerman [16]	1997	Morbidly obese: BMI ≥ 35	Morbidly obese: 84/89 (94.4) Non-obese: 5/89 (5.6)	NR	NR	Morbidly obese: 18 ± 0.7 Non-obese: 7 ± 1.6 ***p* < 0.001**
8	J. E. Varela [8]	2009	NR	Morbidly obese: 62/62 (100)	49 ± 10	IAP ≥ 9 cmH_2_O: 48/62 (77)	NR
9	P. Wacharasint [17]	2016	Underweight: BMI < 18.5 Normal: BMI 18.5–24.9 Overweight: BMI 25–29.9 Obese: BMI ≥30	Underweight: 768/4652 (16.8) Normal: 2624/4652 (57.3) Overweight: 858/4652 (18.7) Obese: 329/4652 (7.2)	NR	Underweight: 6/768 (0.7) Normal: 12/2624 (1.4) Overweight: 20/858 (2.3) Obese: 8/329 (2.4) ***p* = 0.030**	NR

ACS: Abdominal compartment syndrome; BMI: Body mass index; IAH: Intra-abdominal hypertension; IAP: Intra-abdominal pressure; IQR: Inter-quartile range; NR: Not reported; OR: Odds ratio; SD: Standard deviation SAD: Sagittal abdominal diameter. All values of BMI are reported in kg/m^2^. All continuous variables are reported in mean ± standard deviation and all categorical variables are reported in n (%) unless otherwise stated. All values in bold indicate statistical significance, where *p*-value is <0.05. ^*^ Refers to incidence of ACS instead of IAH.

**Table 4 life-13-00330-t004:** Incidence of various co-morbidities in obese versus non-obese patients.

Type of Co-Morbidity	Study Included	Incidence of Co-Morbidity	*p*-Value
Osteoarthropathy	Lambert et al. [11]	Obese: 31/45 (68.9) Non-obese: 0/4 (0)	NR
Gallbladder Disease	Lambert et al. [11]	Obese: 25/45 (55.6) Non-obese: 0/4 (0)	NR
OSA	Lambert et al. [11]	Obese: 25/45 (55.6) Non-obese: 0/4 (0)	NR
GERD	Lambert et al. [11]	Obese: 19/45 (42.2) Non-obese: 0/4 (0)	NR
HTN	Lambert et al. [11]	Obese: 17/45 (37.8) Non-obese: 0/4 (0)	NR
Abdo Hernia	Lambert et al. [11]	Obese: 8/45 (17.8) Non-obese: 0/4 (0)	NR
T2DM	Lambert et al. [11]	Obese: 7/45 (15.6) Non-obese: 0/4 (0)	NR
HLD	Lambert et al. [11]	Obese: 6/45 (13.3) Non-obese: 0/4 (0)	NR
Heart Disease	Lambert et al. [11]	Obese: 5/45 (11.1) Non-obese: 0/4 (0)	NR
Septic Shock	Paolini et al. [12]	Obese: 69/109 (63.3) Non-obese: 210/294 (71.4)	>0.05
Sepsis	Paolini et al. [12]	Obese: 15/109 (13.8) Non-obese: 53/294 (18.0)	>0.05
Cardiovascular Patho	Lambert et al. [11], Paolini et al. [12], Smit et al. [15]	Obese: 27/192 (14.1) Non-obese: 47/346 (13.6)	>0.05
Neurologic Patho	Paolini et al. [12]	Obese: 5/109 (4.6) Non-obese: 28/294 (9.5)	>0.05
Hemorrhagic Shock	Paolini et al. [12]	Obese: 7/109 (6.4) Non-obese: 9/294 (3.1)	>0.05
Trauma	Paolini et al. [12]	Obese: 11/109 (10.1) Non-obese: 44/294 (15.0)	>0.05
ARDS	Paolini et al. [12]	Obese: 3/109 (2.8) Non-obese: 10/294 (3.4)	>0.05
COPD	Smit et al. [15]	Obese: 0/38 (0) Non-obese: 1/148 (0.7)	NR
Immunological Insufficiency	Smit et al. [15]	Obese: 4/38 (10.5) Non-obese: 4/148 (2.7)	**0.034**
Metastasized Neoplasm	Smit et al. [15]	Obese: 1/38 (2.6) Non-obese: 2/148 (1.4)	>0.05
Respiratory Insufficiency	Smit et al. [15]	Obese: 0/38 (0) Non-obese: 1/148 (0.7)	NR
Hematological Malignancy	Smit et al. [15]	Obese: 0/38 (0) Non-obese: 1/148 (0.7)	NR

ARDS: Acute respiratory distress syndrome; COPD: Chronic obstructive pulmonary disease; GERD: Gastro-oesophageal reflux disease; HLD: Hyperlipidemia; HTN: Hypertension; NR: Not reported; OSA: Obstructive sleep apnea; T2DM: Type 2 diabetes mellitus. All categorical variables are expressed in n (%) unless otherwise stated. Values in bold indicate statistical significance, where *p*-value is <0.05.

**Table 5 life-13-00330-t005:** Incidence of various morbidities in obese versus non-obese patients.

Morbidity	Study Included	Patient Demographics	*p*-Value
Need for RRT	Paolini et al. [12]	Obese: 40/109 (36.7) Non-obese: 294 (10.5)	<**0.01**
ICU-acquired infection	Paolini et al. [12], Smit et al. [15], Wacharasint et al. [17]	Obese: 105/476 (22.1) Non-obese: 1124/4592 (24.5)	Wacharasint et al. *p* = **0.047**
Acute coronary syndrome	Paolini et al. [12]	Obese: 6/109 (5.5) Non-obese: 7/294 (2.4)	NR
Thrombophlebitis or PE	Paolini et al. [12]	Obese: 10/109 (9.2) Non-obese: 13/294 (4.4)	NR
New-onset AKI	Smit et al. [15]	Obese: 0/38 (0) Non-obese: 1/148 (0.7)	NR
Length of stay, days	Smit et al. [15]	Obese: 0.94 (5.2) Obese: 0.91 (19.4)	0.060
Need for vasopressors	Smit et al. [15]	Obese: 32/38 (84.2) Non-obese: 132/148 (89.2)	0.571
Reintubation	Smit et al. [15]	Obese: 0/38 (0) Non-obese: 0/148 (0)	NR
SIRS	Wacharasint et al. [17]	Obese: 97/329 (29.5) Non-obese: 1545/4250 (36.4)	**0.001**
ARDS	Wacharasint et al. [17]	Obese: 14/329 (4.3) Non-obese: 159/4250 (3.7)	**0.048**
Mortality	Paolini et al. [12]	Obese: 48/109 (44.0) Non-obese: 78/294 (26.5)	<**0.01**

AKI: Acute kidney injury; ARDS: Acute respiratory distress syndrome; ICU: Intensive care unit; NR: Not reported; PE: Pulmonary embolism; RRT: Renal replacement therapy; SIRS: Systemic inflammatory response syndrome. All categorical variables are expressed as n (%) unless otherwise stated. All continuous variables are expressed as median (range) unless otherwise stated. Values in bold indicate statistical significance, where *p*-value is <0.05.

## Data Availability

Data was extracted from all publicly available published literature. Requests for extracted data may be made with reasonable request to the corresponding author.

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
