# Peer review of "Impact of Obesity on Clinical Outcomes of Patients with Intra-Abdominal Hypertension and Abdominal Compartment Syndrome"

_life, 2023, doi:10.3390/life13020330_

Round 1
Reviewer 1 Report
Thank you for this opportunity to review this manuscript. The authors conduct a metanalysis of 9 studies to increase power to prove their hypothesis that IAH is increased in obesity which is an independent risk factor for ACS. They also demonstrate that other markers of morbidity and disease processes are increased as well. Over allt his is a well written paper and while the conclusions are likely not new, the powering of a metanalysis such as this does add to the body of literature as the average weights of patients in developed nations continues to increase.
Abstract: Overall this is well written
Introduction:
Overall, the overview is excellent. However, the reader who is not used to ICU management may benefit from a review of how IAP is measures (bladder pressures, peak airway pressures etc) and how ACS is treated, as laparotomies for ACS are not negligible mortality/operative risk.
I would also mention pancreatitis as a primary cause, as it is far more common than AAA rupture or RP hematoma.
Materials and Methods
43 papers were excluded as “other”. A brief explanation into this should be included.
If BMI was not used for the other 5 papers, what was the definition of obesity?
Results
While I understand why the definition of BMI may have been disregarded and a binary “yes/no” for whether the authors thought the patients was obese was used, this creates potential confounding as the average BMI for obesity in this study is somewhere between 30 and 35. This should at least be included as a limitation. Considering this, the results table may need clarification, particularly under “BMI classification” for studies 4, 5, and 8.
Discussion
Overall, this is a good review. It could be condensed as it reads quite long, particularly with regards to obesity complications that fall outside of the stated goals of the hypothesis.
Reviewer 2 Report
This paper reviewed the papers on the symptom (Intra-Abdominal Hypertension) and well organized them for easy understanding. Although there is no paper directly comparing abdominal compartment syndrome between obese and non-obese, the authors have been well supplemented using IAH symptoms. In conclusion, I recommend accepting this review paper without major revision comments.
Reviewer 3 Report
The manuscript represents an interesting systematic review that uses PRISMA, on two very important topics (intra-abdominal hypertension and abdominal compartment syndrome).
The introductory part should be developed and some more relevant references added. Aspects related to the history of the notions, the values from which we speak of IAH and ACS and the measurement method should be mentioned.
Socea B., Nica A.A., Smaranda A., Bratu O.G., Diaconu C.C., Carap A.C., Neagu T.P., Badiu C.D., Constantin V.D. Abdominal Compartment Syndrome—A Surgical Emergency. Mod. Med. 2018;25:187–191. doi: 10.31689/rmm.2018.25.4.187.
Păduraru DN, Andronic O, Mușat F, Bolocan A, Dumitrașcu MC, Ion D. Abdominal Compartment Syndrome-When Is Surgical Decompression Needed? Diagnostics (Basel). 2021 Dec 7;11(12):2294. doi: 10.3390/diagnostics11122294.
